# Hypothalamic NPY-Y1R Interacts with Gonadal Hormones in Protecting Female Mice against Obesity and Neuroinflammation

**DOI:** 10.3390/ijms23116351

**Published:** 2022-06-06

**Authors:** Alessandra Oberto, Ilaria Bertocchi, Angela Longo, Sara Bonzano, Silvia Paterlini, Clara Meda, Sara Della Torre, Paola Palanza, Adriana Maggi, Carola Eva

**Affiliations:** 1Neuroscience Institute of the Cavalieri-Ottolenghi Foundation, Regione Gonzole 10, Orbassano, 10043 Turin, Italy; alessandra.oberto@unito.it (A.O.); ilaria.bertocchi@unito.it (I.B.); angela.longo@unito.it (A.L.); sara.bonzano@unito.it (S.B.); 2Department of Neuroscience, University of Turin, C.so Massimo d’Azeglio 52, 10126 Turin, Italy; 3Neuroscience Institute of Turin, Regione Gonzole 10, Orbassano, 10043 Turin, Italy; 4Department of Life Sciences and Systems Biology (DBIOS), University of Turin, Via Accademia Albertina 13, 10123 Turin, Italy; 5Department of Medicine and Surgery, University of Parma, Behavioral Biology Lab—Parco Area delle Scienze 11A, 43124 Parma, Italy; silvia.paterlini@unipr.it (S.P.); paola.palanza@unipr.it (P.P.); 6Department of Pharmaceutical Sciences, University of Milan, Via Balzaretti 9, 20133 Milan, Italy; clara.meda@unimi.it (C.M.); sara.dellatorre@unimi.it (S.D.T.); adriana.maggi@unimi.it (A.M.); 7Center of Excellence on Neurodegenerative Diseases, University of Milan, Via Balzaretti 9, 20133 Milan, Italy

**Keywords:** obesity, high-fat diet, ovariectomy, hypothalamic Y1R receptors, neuroinflammation

## Abstract

We previously demonstrated that Npy1r*^rfb^* mice, which carry the conditional inactivation of the *Npy1r* gene in forebrain principal neurons, display a sexually dimorphic phenotype, with male mice showing metabolic, hormonal and behavioral effects and females being only marginally affected. Moreover, exposure of Npy1r*^rfb^* male mice to a high-fat diet (HFD) increased body weight growth, adipose tissue, blood glucose levels and caloric intake compared to Npy1r*^2lox^* male controls. We used conditional knockout Npy1r*^rfb^* and Npy1r*^2lox^* control mice to examine whether forebrain disruption of the *Npy1r* gene affects susceptibility to obesity and associated disorders of cycling and ovariectomized (ovx) female mice in a standard diet (SD) regimen or exposed to an HFD for 3 months. The conditional deletion of the *Npy1r* gene increased body weight and subcutaneous white adipose tissue weight in both SD- and HFD-fed ovx females but not in cycling females. Moreover, compared with ovx control females on the same diet regimen, Npy1r*^rfb^* females displayed increased microglia number and activation, increased expression of Neuropeptide Y (NPY)-immunoreactivity (IR) and decreased expression of proopiomelanocortin-IR in the hypothalamic arcuate nucleus (ARC). These results suggest that in the ARC NPY-Y1R reduces the susceptibility to obesity of female mice with low levels of gonadal hormones and that this effect may be mediated via NPY-Y1R ability to protect the brain against neuroinflammation.

## 1. Introduction

The incidence of metabolic disorders including obesity in women is lower than in men; however, prevalence of these disorders increases dramatically after menopause and is associated with an increased risk of inflammatory disorders (including type II diabetes, cardiovascular diseases) [1,2,3]. Considering the well described anti-inflammatory role of estrogens [4], it is conceivable that there is a role for this hormone in these dysmetabolic disorders.

A sexually dimorphic vulnerability to obesity is generally observed in mammals. For instance, in rodents exposed to a high-fat diet (HFD), females are generally more resistant to HFD-induced obesity than males [5,6,7,8]. Most interestingly, in male mice HFD-induced obesity is associated with hypothalamic microglial activation [9,10], while this is not observed in females that do not show the HFD-induced accumulation of reactive, pro-inflammatory microglia in the hypothalamus [11,12]. This suggests an involvement of the hypothalamus in the differential susceptibility to HFD-induced obesity of males and females.

Estradiol plays a leading role in the regulation of energy homeostasis in female mammals, and declining levels of estradiol in ovariectomized (ovx) rodents and postmenopausal women are associated with body weight increase and fat redistribution [2,13,14,15]. Moreover, estradiol has a protective effect against the development of diet-induced microgliosis in the hypothalamic nuclei involved in the control of food intake [5,10,11]. Several studies have demonstrated that, at the hypothalamic level, estrogens inhibit food intake and adipocyte activity by decreasing the excitability of the neuropeptide Y (NPY)/Agouti-related protein (AgRP) neurons in the arcuate (ARC) [15,16,17], which project to other nuclei of the circuit controlling food intake, including the paraventricular (PVN), ventromedial (VMH) and dorsomedial (DMH) nuclei [18]. Estrogens modulate NPY expression and *Npy1r* gene activity [15,19] and regulate Y1 (Y1R) expression during the estrous cycle [20] or after exposure to different diets [21], thus playing a role in the changes in feeding behavior characteristic of the estrous cycle. NPY is identified as the most potent orexigenic peptide that responds to peripheral orexigenic and anorexigenic signals such as ghrelin, leptin and insulin to regulate feeding [22,23,24,25]. Acute administration of NPY increases food intake and induces hyperinsulinemia [26], whereas its chronic administration produces hyperphagia and obesity and decreases thermogenesis [27]. Dysfunctions of the NPY system have been observed in diseases such as obesity, type II diabetes and metabolic syndrome [28,29,30]. NPY interacts with a family of G-protein coupled receptors, the most well-known being the Y1, the Y5 and the presynaptic Y2 receptors [31,32,33]. 

Several studies demonstrated that the NPY-Y1R system is sexually dimorphic and sensitive to gonadal steroids [16,17,32,34]. Sex-specific differences in NPY and Y1R expression in the hypothalamic system controlling food intake have been found by our and others’ laboratories [35,36,37,38,39]. Conditional knock-out mice (Npy1r*^rfb^* mice), carrying the deletion of the *Npy1r* gene in in forebrain principal neurons starting from a juvenile age [40,41], show a sexually dimorphic phenotype, revealing the existence of NPY-Y1R neuronal pathways involved in sex-biased differences of metabolic functions [42]. Indeed, Npy1r*^rfb^* male mice exhibited hyperactivation of the hypothalamus–pituitary–adrenal (HPA) axis, reduced body weight growth, after-fasting refeeding, white adipose tissue (WAT) and plasma leptin levels compared with their Npy1r*^2lox^* control littermates. Conversely, mutant female mice appeared to be resilient to the effects of *Npy1r* gene inactivation on hormone and metabolic functions [42]. Interestingly, *Npy1r* mRNA expression was reduced in the ARC and in the PVN of female, but not male mice. In addition, Npy1r*^rfb^* male mice fed with an HFD display increased body weight, visceral adipose tissue and blood glucose levels, and food and calories intake as compared to control Npy1r*^2lox^* mice, suggesting that, in male mice, inactivation of the *Npy1r* gene increases susceptibility to diet-induced obesity and glucose intolerance, which may be indexes of metabolic syndrome [43].

To better understand the role of gonadal hormones–NPY system interaction and hypothalamic microglia activity in the sex-specific susceptibility to obesity and associated disorders, in the present study we investigated the effect of the conditional inactivation of the *Npy1r* gene in the forebrain of sham (correctly cycling) and ovariectomized (ovx) female mice fed with a standard diet (SD) or with an HFD.

## 2. Results

Table 1 summarizes the ANOVA analysis of the results.

### 2.1. Conditional Inactivation of the Npy1r Gene Is Associated with Increased Body and Fat Weight in Ovariectomized, but Not in Cycling Female Mice

#### 2.1.1. Body Weight

After the beginning of the comparison of SD and HFD effects (day 60) in female mice, periodic measurement of body weight did not show significant differences between cycling Npy1r*^2lox^* and Npy1r*^rfb^* female mice, independently of the diet regimen (Figure 1A). This was not the case in ovx mice, where the conditional ablation of the *Npy1r* gene induced a growth in body weight that in time became significantly more pronounced in the Npy1r*^rfb^* mice. Furthermore, mutant mice appeared to be also more susceptible to the effect of the HFD, with a generalized trend to an increased weight growth which was significant towards the middle of the treatment (Figure 1B). The lack of significant differences after the 60th day of the diet regimen (postnatal day 120, P120) between HFD-fed ovx Npy1r*^2lox^* and Npy1r*^rfb^* females was attributed to the presence of a ceiling effect. 

These observations suggested that gonadal hormones were able to functionally mask the lack of Y1R activity in glutamatergic neurons, thus maintaining a balanced body weight in the SD and a physiological response to the HFD. 

#### 2.1.2. White Adipose Tissue (WAT) Weight

Overall, the HFD and ovariectomy significantly increased subcutaneous, visceral and gonadic WAT weight (Figure 1C). Similar to what was observed with regard to body weight, both SD- and HFD-fed ovx Npy1r*^rfb^* females displayed greater subcutaneous WAT weight compared with ovx control female mice on the same diet regimen (Figure 1C). No significant differences in visceral and gonadic WAT weight were observed between cycling or ovx Npy1r*^2lox^* and Npy1r*^rfb^* female mice, independently of the diet regimen (Figure 1C).

#### 2.1.3. Leptin Plasma Levels

Both the HFD and ovariectomy significantly increased leptin plasma levels (Figure 1D). Neither cycling nor ovx Npy1r*^rfb^* females showed differences in leptin plasma levels compared with their control littermates on the same diet regimen (Figure 1D). These results suggest that an altered peripheral regulation of leptin release in the absence of gonadal hormones might determine the lack of enhanced leptin levels despite the increase of WAT weight in ovx Npy1*^rfb^* mice. 

### 2.2. Npy1r Gene Inactivation Decreased Locomotor Activity but Failed to Affect the Food Intake of SD- and HFD-Fed Cycling and Ovariectomized Female Mice

#### 2.2.1. Spontaneous Locomotor Activity

Overall, the HFD and ovariectomy decreased spontaneous locomotor activity in all experimental groups (Figure 2A). *Npy1r* gene inactivation further decreased spontaneous locomotor activity in HFD-fed cycling females and in SD-fed ovx females (Figure 2A). To test the relationship between body weight and locomotor activity, we used a regression analysis with locomotor activity as the dependent variable and body weight as the predictor. Body weight statistically predicted locomotor activity (b = −929.52, t(86) = −7.74, *p* < 0.001) and explained 64% of the variance in locomotor activity (R2 = 0.64, F(1,86) = 60.20, *p* < 0.001) (Figure 2B).

#### 2.2.2. Daily Food Intake

Analysis of the average cumulative amount of chow eaten showed that female mice of the different experimental groups consumed the same total amount of food over the entire length of the experiment, independently of the genotype (Figure 2C). Accordingly, the cumulative energy intake of HFD-fed cycling and ovx females was significantly higher than that measured in the corresponding controls on the SD regimen (Figure 2D). These data are confirmed by the analysis of energy intake every 9 days in the eight groups of mice. Both cycling and ovx control and conditional mutant mice increased energy intake when fed with HFD and no differences were observed during the length of the experiment (Figure 2E). 

These results suggest that a reduced locomotor activity rather than an increased caloric intake might contribute to determine the obesity-like phenotype of Npy1r*^rfb^* conditional mutants in the absence of gonadal hormones. 

### 2.3. Npy1r Gene Inactivation Failed to Affect Glucose Homeostasis and Metabolism, Heart Rate and Blood Pressure of SD- and HFD-Fed Cycling and Ovariectomized Female Mice 

#### 2.3.1. Glucose Homeostasis and Metabolism

Glucose tolerance tests (GTT, to assess glucose clearance from the blood) and insulin tolerance tests (ITT, to assess insulin resistance) were performed as measures of glucose homeostasis.

Three-way ANOVA analysis for repeated measures of glucose concentrations during GTT revealed that the HFD and ovariectomy decreased glucose tolerance during GTT (Figure 3A). Moreover, both ovariectomy and diet significantly increased the area under the curve (AUC) (Figure 3B).

In addition, both cycling and ovx HFD-fed mice showed delayed glucose clearance in response to exogenous insulin during ITT (Figure 3C). HFD also increased the AUC (Figure 3D).

To further explore the impact of ovariectomy and an HFD on liver glucose metabolism in control and mutant female mice, we examined pyruvate incorporation into glucose via gluconeogenesis in a pyruvate tolerance test (PTT). As shown in Figure 3E, the HFD increased glucose concentrations during PTT in cycling but not in ovx mice. Moreover, HFD-fed cycling females showed a significant increase of the AUC (Figure 3F). 

Neither cycling nor ovx Npy1r*^rfb^* females showed differences in GTT, ITT and PTT compared with their control littermates on the same diet regimen (Figure 3A–F).

Taken together, these results show an increased adiposity without significant changes in glucose homeostasis and metabolism in ovx Npy1r*^rfb^* females and suggest a specific role of Y1R in the regulation of fat accumulation in the absence of gonadal hormones.

#### 2.3.2. Heart Rate and Blood Pressure

A significant increase of heart rate was observed in ovx females, independently of genotype and diet regimen (Figure 3G). No difference in blood pressure was observed among mice groups (Figure 3H).

### 2.4. Effects of Npy1r Inactivation on the Expression of Neuropeptides Implicated in the Control of Energy Metabolism in the ARC and PVN of SD- and HFD-Fed Cycling and Ovariectomized Female Mice

#### 2.4.1. AgRP

Overall, both ovx and HFD-fed mice showed a significant decrease of AgRP expression in the ARC compared with cycling and SD-fed females, respectively (Figure 4A). Conditional inactivation of the *Npy1r* gene induced a significant decrease of AgRP-IR in the ARC of SD-fed cycling females compared to their control littermates. A decrease of AgRP-IR was also observed in SD-fed ovx Npy1r*^rfb^* females, although it was not significant (Figure 4A,C). 

A similar pattern of AgRP expression was observed in the PVN. The overall analysis revealed a significant effect of ovariectomy, diet and genotype (Figure 4B). 

#### 2.4.2. NPY

As shown in Figure 4D, SD-fed ovx Npy1r*^2lox^* females displayed a significant decrease of NPY expression compared with SD-fed cycling control females. In ovx females, the inactivation of the *Npy1r* gene significantly increased NPY-IR in the ARC of Npy1r*^rfb^* mice compared with Npy1r*^2lox^* mice on the same diet regimen (Figure 4D,F). No significant differences in NPY expression were observed among the experimental groups in the PVN (Figure 4E).

#### 2.4.3. Proopiomelanocortin (POMC)

As shown in Figure 4G,I, a significant decrease of POMC-IR was observed in the ARC of SD-fed ovx Npy1r*^rfb^* females compared with their corresponding controls. In the PVN, both ovariectomy and diet significantly decreased POMC-IR, independently of the genotype (Figure 4H).

### 2.5. Npy1r Gene Inactivation Is Associated with Increased Microglial Number and Activation in the Arcuate Nucleus of Ovariectomized, but Not Cycling Female Mice

The total cell body size divided by the total cell size (cb/c) was used as measure of microglial activation. The genotype did not affect microglial activation (Figure 5A) and the number of microglia (Figure 5C) in cycling animals. The *Npy1r* gene deletion clearly affected the microglia state of activity after ovariectomy as both the morphology (Figure 5B) and number of microglia cells (Figure 5D) were increased in animals fed by the SD and even more in animals subjected to the HFD (Figure 5E,F).

These data pointed to a direct relationship of microglia activity in the ARC with the effects of ovariectomy and diet on body weight.

## 3. Discussion

In this study we investigated the role of Y1R on vulnerability to diet-induced obesity and associated disorders of cycling and ovx female mice. To this aim, we used a conditional knockout mouse model in which the postnatal inactivation of the *Npy1r* gene in glutamate-containing neurons differentially affects male and female phenotypes, with Npy1r*^rfb^* male mice showing metabolic, hormonal and behavioral effects and Npy1r*^rfb^* females being only marginally affected [42,43].

Here, we demonstrated that the conditional deletion of the *Npy1r* gene did not affect body weight and WAT weight in HFD-fed cycling female mice, thus confirming sex-related effects of NPY-Y1R functions. However, a clear effect of the conditional deletion of the *Npy1r* gene was observed after ablation of the ovaries; indeed, in ovx mice we observed the mutants showing an increased body and WAT weight in mice fed with an SD and such an effect was enhanced by the HFD regimen. These findings suggest that Y1R might interact with gonadal hormones in counteracting the development of obesity. The ARC neuroinflammatory response observed in the conditional Npy1r*^rfb^* females after ovariectomy further supports a central role of gonadal hormones in the maintenance of a homeostatic control.

However, as suggested by the data on systemic leptin levels, the peripheral impact of the cross-talk mechanisms linking Y1R to gonadal hormones could be not always detectable. In fact, the HFD-induced increase in WAT weight was associated with increased leptin circulating levels and the conditional deletion of the *Npy1r* gene in cycling females did not affect either WAT weight or leptin release, thus confirming that the increase in leptin concentration is in proportion to the size of white fat depots [44]. Leptin synthesis is known to also be modulated by sex hormones, specifically suppressed by testosterone and increased by estrogen and progesterone [45]. Here, despite a further increase in WAT weight recorded in Npy1r*^rfb^* ovx female, no significant changes in level of this adipokine was detected, thus indicating a prevalence of sex-related control of leptin secretion, at least in our experimental conditions. On the other hand, we cannot exclude the involvement of other adipokines or gastric hormones, such as ghrelin, that induces adiposity independently of the orexigenic effect [46].

In cycling females, HFD exposure increased energy intake, body weight growth, subcutaneous WAT, plasma leptin levels and decreased glucose tolerance in both control and conditional mutant females. These metabolic alterations were associated with a concomitant decrease of locomotor activity with no changes in feeding behavior. Moreover, HFD exposure decreased AgRP-IR in the ARC of Npy1r*^2lox^* and Npy1r*^rfb^* cycling females, a decrease that was also observed in SD-fed Npy1r*^rfb^* mutants, as previously reported [42]. These results are in agreement with previous studies showing that ARC AgRP neurons are very sensitive to peripheral metabolic signals [47,48,49,50] and suggest that the decreased AgRP expression in the ARC may be a consequence of the increase of WAT mass and plasma leptin levels. Conversely, no significant differences were observed in NPY expression in the ARC of HFD-fed Npy1r*^2lox^* and Npy1r*^rfb^* cycling females compared with cycling females on the SD regimen. The different response to the HFD of AgRP- and NPY-IR in cycling females could be explained by the observation that, in the ARC, leptin receptors are expressed in a small population of NPY neurons and that the NPY neurons expressing leptin receptors do not co-express AgRP [23,25]. 

The most significant finding of this study was the observation that both SD- and HFD-fed ovx Npy1r*^rfb^* females showed a significant increase of body weight growth and WAT weight that is associated with microglia activation and proliferation in the ARC. 

A number of studies have shown that exposure to HFD promotes microgliosis and increases the release of pro-inflammatory cytokines mostly in the ARC [9,10,51,52], which contains both anorexigenic and orexigenic neurons involved in the neural control of energy homeostasis.

This rapid HFD-induced pro-inflammatory response precedes the HFD-induced weight gain, suggesting that neuroimmune response in the ARC might play a causal role in the pathophysiology of obesity [51]. Previous studies demonstrated that sex differences in microglial activation in the brain nuclei involved in the modulation of energy homeostasis and estradiol have protective effects against HFD-induced microgliosis in the hypothalamus [11,12,53]. Accordingly, here we report that the HFD increased microglial number and activation in the ARC of ovx, but not of cycling Npy1r*^rfb^* females, thus pointing to a role of NPY in the sex-specific control of energy homeostasis. 

Emerging evidence has indicated that NPY acts as a critical link between the nervous system and immune system and demonstrated that NPY directly modulates immune cells by acting on NPY receptors [54]. Microglia expresses Y1R and NPY, acting through Y1R, and significantly inhibits microglial activation, phagocytosis and cytokine secretion [55,56,57,58]. In Npy1r*^rfb^* mice, the conditional inactivation of the *Npy1r* gene is driven by the alpha-CaMKII promoter, which is expressed specifically in forebrain glutamate positive cells [41]. Given that microglia express alpha-CamKII [59,60], we hypothesize that, in the absence of the inhibitory role of gonadal hormones on neuroinflammation, the conditional deletion of the *Npy1r* gene in microglia results in the activation and proliferation of microglia and, in turn, in the obesity observed in SD- and HFD-fed Npy1r*^rfb^* ovx females. In the present study, we did not carry out hormone replacement; thus, our experimental design does not allow the specific impact of each ovarian steroid on body weight increase and microglia activation on ovx females to be defined. However, estradiol emerged as a leading candidate since progesterone replacement does nor revert ovariectomy-induced obesity [61] and progesterone receptors are not expressed in microglia in adult mice [62] (Figure 6). 

On the other hand, on the basis of previous studies [15,19,20], we cannot exclude that other cross-talk mechanisms between estrogen and NPY may exist in neurons and glia. 

It is well known that hypothalamic inflammation reduces the responsiveness of POMC and NPY neurons in the ARC to the physiological actions of leptin and insulin and alters the normal activity of the orexigenic and anorexigenic peptides, thereby promoting energy imbalance and obesity [48]. Here we demonstrated that the inflammatory response and the metabolic alterations observed in Npy1r*^rfb^* ovx females are associated with an increased expression of NPY in the ARC. Moreover, a significant decrease of POMC-IR was observed in the ARC of ovx Npy1r*^rfb^* females in SD regimen. These findings are in line with previous studies showing that neuroinflammation is associated with an alteration of NPY and POMC neuron activity [50,63]. 

In conclusion, here we provide the first evidence that the NPY-Y1R transmission in the ARC plays a pivotal role in reducing susceptibility to obesity in female mice with low levels of ovarian hormones and that this protective effect may be mediated via NPY ability to protect the brain against neuroinflammation. These results may provide a better understanding of the causative mechanisms for the multimorbidities associated with females’ reproductive aging, including the metabolic syndrome, cognitive decline and neurodegenerative disorders.

## 4. Materials and Methods

### 4.1. Animals

Npy1r*^2lox^* and Npy1r*^rfb^* B6129S (129/SvJ, C57BL/6N derived strain) female mice were generated as previously described [41]. Mice were housed in a temperature (22 ± 1 °C) and humidity (50 ± 10%) controlled room, in groups of 4–6 per cage, with ad libitum access to food and water. Nesting paper materials and cardboard houses were used for environmental enrichment and diurnal rhythm was maintained with a 12:12 h light–dark cycle (08:00 a.m.–08:00 p.m.).

Eight-week-old (P48) Npy1r*^2lox^* and Npy1r*^rfb^* female mice were sham operated (cycling) or ovx and maintained on an SD. Starting from postnatal day 60 (P60), ovx and cycling females were fed a hypercaloric HFD or maintained on an SD until the sacrifice.

All cycling females were in estrous cycle at euthanasia that was induced by anesthesia overdosing (ketamine and xylazine, Vetefarma, Cuneo, Italy) and cervical dislocation. Vaginal smears were performed at 9:00–10:00 a.m. Mice were euthanized at 5–5.5 months of age between 2:00 and 5:00 p.m.

All experiments were conducted in accordance with the European Community Council Directive of 24 November 1986 (86/EEC) and approved by the University of Turin Ethical Committee for animal research and by the Italian Ministry of Health (licenses no. 574/2016).

### 4.2. Body Weight and Food Intake Analysis

Mice body weight and food intake were measured twice a week from P60 to P156 with a digital scale accurate to 0.01 g, always from 10:00 a.m. to 11:00 a.m. Food intake was quantified by calculating the difference between pre- and post-weighed food. The amount of food consumed by the single cage was averaged by the number of mice and the number of days, then the Kcal consumed were calculated for each diet.

### 4.3. Diet Composition

The SD consisted of 4.3% fat, 67.3% carbohydrate and 19.2% protein with 10% Kcal derived from fat, 70% Kcal derived from carbohydrates and 20% Kcal derived from proteins (D12450B, Research Diets, NJ, USA). The HFD consisted of 35% fat, 26% carbohydrate and 26% protein with 60% of Kcal derived from fat, 20% Kcal from carbohydrates and 20% Kcal from proteins (D12492, Research Diets, NJ, USA). Free access to the diets was provided. 

### 4.4. Spontaneous Locomotor Activity

Spontaneous locomotor activity in the home cage was performed on females in estrous cycle. Locomotor activity was continuously recorded for 3 h using an infrared video camera, starting at the onset of the dark phase (active period [42]). Mice were allowed to become accustomed to their clean home cage for 24 h before starting collection of the data. The total distance (meters) traveled by mice was recorded and data were analyzed automatically from the digitized images by using a computerized video tracking software (Ethovision XT 15 video track system; Noldus Information Technology, Wageningen, The Netherlands).

### 4.5. Glucose (GTT), Insulin (ITT) and Pyruvate (PTT) Tolerance Tests

GTT and PTT were performed following an overnight fasting (12 h), while for ITT mice were fasted for 5 h prior to the test. All tests were performed between 9:00 a.m. and 11:00 a.m. Blood glucose levels from tail bleeding were measured using a glucose meter FreeStyle Lite (Abbot, Alameda, CA, USA) at 0, 30, 60, 90 and 120 min after i.p. injection of a 1 g/kg d-glucose solution for GTT, a 0.5 g/kg insulin solution for ITT and a 1 g/kg pyruvate solution for PTT (Carlo Erba Reagent S.R.L., Milan, Italy). GTT, ITT and PTT were performed starting on P161 and one-week time was allowed for the mice to recover between tests. 

### 4.6. Heart Rate and Blood Pressure

Blood pressure and heart rate were measured by tail cuff plethysmography by using a BP-2000 Series II Blood Pressure Analysis System, 2 channels mouse platform (Visitech Systems, Napa Pl, Apex, NC, USA) at P160 as previously described [64]. 

### 4.7. Tissue Collection and Analysis

Blood samples were collected from the atrium of anesthetized mice. EDTA (12.5 mM, Merck, Life Science, Milano, Italy) was added immediately to prevent clotting and plasma, obtained by centrifugation at 8000 rpm for 10 min at 4 °C, was stored at −80 °C until analysis. Levels of circulating leptin were measured in duplicate with a commercially available kit (Mouse leptin ELISA, Merck, Life Science, Milano, Italy). To avoid the inter-assay variability, all samples were run in a single assay. WAT (subcutaneous, visceral and gonadic) was collected and weighed, then frozen on dry ice for later analysis.

### 4.8. Histological Analysis

NPY, AgRP, POMC and ionized calcium-binding adaptor protein-1 (IBA-1) immunohistochemical analysis was performed on postfixed brains. Right after blood collection, mice brains were dissected and fixed for 24 h with 4% paraformaldehyde in PBS. Free-floating cryostat brain coronal sections (30 μm) were immunostained and DAB-developed as previously described [42]. For each peptide analyzed, sections were incubated for 2 days with 1:4000 rabbit anti-POMC, 1:3000 rabbit anti-NPY, 1:2500 rabbit anti-AgRP (all three from Phoenix Europe GMBH, Karlsruhe, Germany) or 1:2000 rabbit-anti IBA-1 (Wako Chemicals—Richmond, VA, USA) primary antibody.

NPY, POMC and AgRP quantitative analysis was performed on three to five sections of ARC (between −1.58 and −1.82 mm relative to bregma) and PVNmp (−0.94 to −0.82 mm relative to bregma) from each animal by one operator blind to the genotype, treatment and diet. Brain slices images (2088 × 1550 pixels) were captured using the 10× objective of a Zeiss Axioplan microscope equipped with a digital camera (Leica DFC 320, Wetzlar, Germany). ImageJ software (NIH, Bethesda, MD, USA) was used to measure immunoreactivity through a threshold method and the number of positive pixels and the extension of area of interest were used to determine the fractional area covered by the specific signal.

### 4.9. Analysis of Microglial Activation

To quantify the number of microglial cells in the ARC and their suggestive activation state, 30 μm slices immunostained for IBA-1 were scanned at 20× magnification on a Zeiss Axioscan 7, High-performance Slide Scanner (Carl-Zeiss, Oberkochen, Germany) and three to five sections of ARC (between −1.58 and −1.82 mm relative to bregma) for each animal were analyzed using ImageJ software as described before [65]. The number of microglia were counted and corrected for the size of the area of interest. Cell body to total cell size ratio (cb/c) was determined and utilized as a measurement for morphological changes that suggest microglial activation. 

To generate cb/c, the “Adjusted threshold” and “Analyze particles” functions of ImageJ software were applied to 8-bit images of ARC. The intensity threshold (default algorithm) and size filter were applied to measure the pixels from the cell body size and pixels from the total cell size (cell body + dendrites) as previously described [65,66]. Based on area coverage, the total microglial cell size and total microglial cell body size were determined. The total cell body size was divided by the total cell size to obtain the cb/c as a measure of microglial activation. 

### 4.10. Statistical Analysis

A three-way ANOVA (genotype, diet and ovariectomy) for repeated measures (days and minutes, RM) was run to analyze body weight, calories intake and GTT, ITT and PTT during metabolic challenge. For mean body weight, this analysis was followed by a two-way ANOVA for repeated measures (genotype and diet as independent variables, days as repeated measures) independently performed on cycling and ovx females. A three-way ANOVA (genotype, diet and ovariectomy) was run to analyze WAT, plasma leptin levels, spontaneous locomotor activity, food and calories intake, AUC, heart beats, systolic pressure, AgRP-, NPY-, POMC-IR and IBA-1 IR. For microglia cb/c and microglia number, this analysis was followed by a 2-way ANOVA (genotype and diet as independent variables) independently performed on cycling and ovx females. The association between body weight and locomotor activity was analyzed by a regression analysis with locomotor activity as the dependent variable and body weight as the predictor. Data were analyzed by means of Statistica 10.0 (Stat-Soft, Tulsa, OK, USA). The significance level was set at *p* < 0.05. Post hoc tests were performed by the Newman–Keuls test. All data are expressed as mean ± SEM, and the level of statistical significance was set at *p* < 0.05. Symbols used to indicate statistical significance are: & for the main factor effects of the ANOVA and * (genotype), # (diet) and § (ovariectomy) for the post hoc tests. 

## Figures and Tables

**Figure 1 ijms-23-06351-f001:**
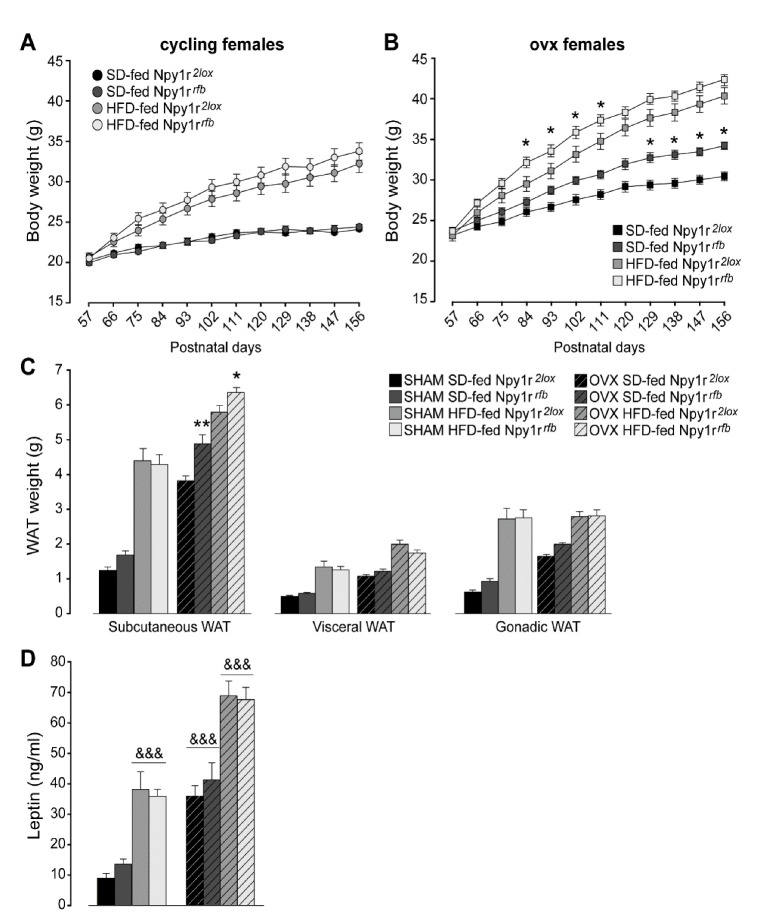
Body weight of standard diet (SD)- and high-fat diet (HFD)-fed cycling (**A**) and ovx (**B**) Npy1r*^2lox^* control and Npy1r*^rfb^* knockout female mice. No significant differences in body weight growth were observed between cycling Npy1r*^2lox^* and Npy1r*^rfb^* female mice during postnatal days (P) 57–156, independently of diet regimen. SD- and HFD-fed ovx Npy1r*^rfb^* females displayed a significantly greater body weight compared with their control littermates on the same diet regimen. Data are the mean ± SEM; n = 17–26 from 6 litters. * *p* < 0.05 versus the corresponding control females. (**C**) Adiposity of standard diet (SD)- and high-fat diet (HFD)-fed cycling and ovx Npy1r*^2lox^* control and Npy1r*^rfb^* knockout female mice. Significantly larger depots of subcutaneous fat were observed in ovx Npy1r*^rfb^* females compared with their control littermates on the same diet regimen. Data are the mean ± SEM; n = 10–15 from 3 litters. ** *p* < 0.01 and * *p* < 0.05 versus SD-fed and HFD-fed ovx Npy1r*^2lox^* female mice, respectively. (**D**) Leptin plasma levels of standard diet (SD)- and high-fat diet (HFD)-fed cycling and ovx Npy1r*^2lox^* control and Npy1r*^rfb^* knockout female mice. Both ovariectomy and HFD increased leptin plasma levels of female mice. Data are the mean ± SEM; n = 10–15 from 3 litters. &&& *p* < 0.001.

**Figure 2 ijms-23-06351-f002:**
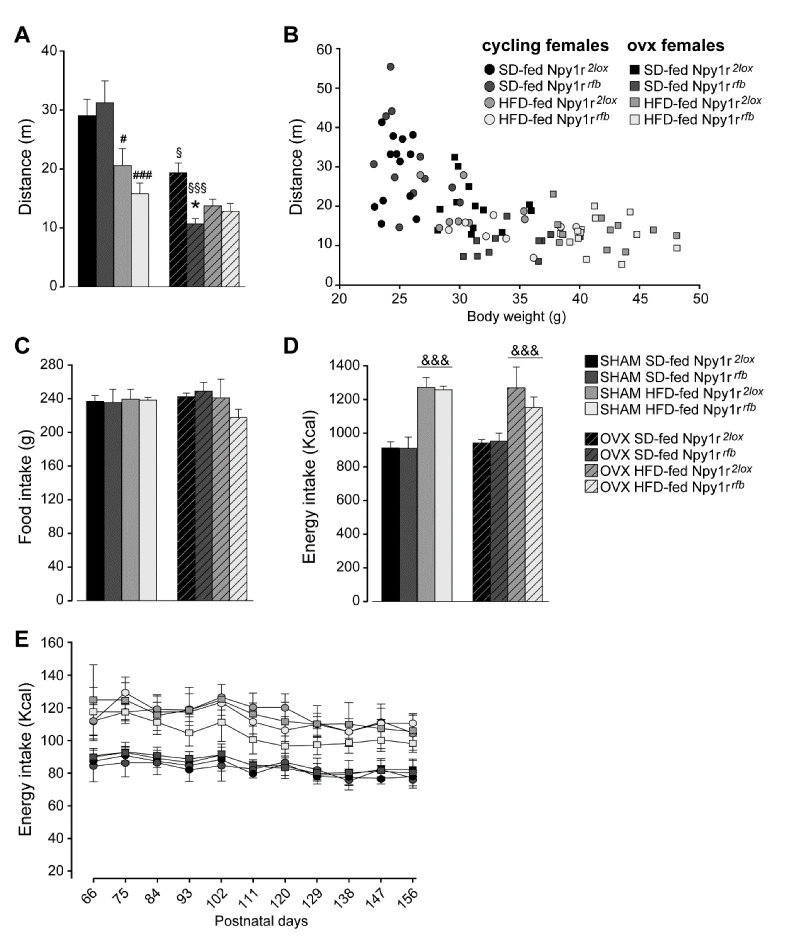
(**A**) Spontaneous locomotor activity of standard diet (SD)- and high-fat diet (HFD)-fed cycling and ovx Npy1r*^2lox^* control and Npy1r*^rfb^* knockout female mice. Both HFD and ovariectomy significantly decreased locomotor activity (distance traveled, cm) of Npy1r*^2lox^* and Npy1r*^rfb^* female mice. HFD-fed cycling and SD-fed ovx Npy1r*^rfb^* females showed significantly lower locomotor activity compared with their corresponding controls. Data are the mean ± SEM; n = 8–12 from 2 litters. # *p* < 0.05 versus SD-fed cycling controls; ### *p* < 0.001 versus SD-fed cycling Npy1r*^rfb^* female mice; § *p* < 0.05 versus SD-fed cycling controls; §§§ versus SD-fed cycling Npy1r*^rfb^* female mice; * *p* < 0.05 versus SD-fed ovx controls. (**B**) Correlation between locomotor activity and body weight of 156 days old SD- and HFD-fed cycling and ovx Npy1r*^2lox^* and Npy1r*^rfb^* female mice. Data are the mean ± SEM; n = 6 from 3 litters. Food and energy intake of standard diet (SD)- and high-fat diet (HFD)-fed cycling and ovx Npy1r*^2lox^* control and Npy1r*^rfb^* knockout female mice (**C**–**E**). (**C**) No significant differences of cumulative food intake (g) were observed among the different experimental groups. Data are the mean ± SEM; n = 4 from 3 litters. (**D**) HFD increased cumulative energy intake (Kcal) of cycling and ovx female mice, independently of the genotype. Data are the mean ± SEM; n = 6 from 3 litters. &&& *p* < 0.001 versus SD-females. (**E**) Nine-day cumulative energy intake (kcal) of SD- and HFD-fed cycling and ovx Npy1r*^2lox^* control and Npy1r*^rfb^* knockout female mice during the metabolic challenge procedure (P66–156). Data are the mean ± SEM; n = 6 from 3 litters.

**Figure 3 ijms-23-06351-f003:**
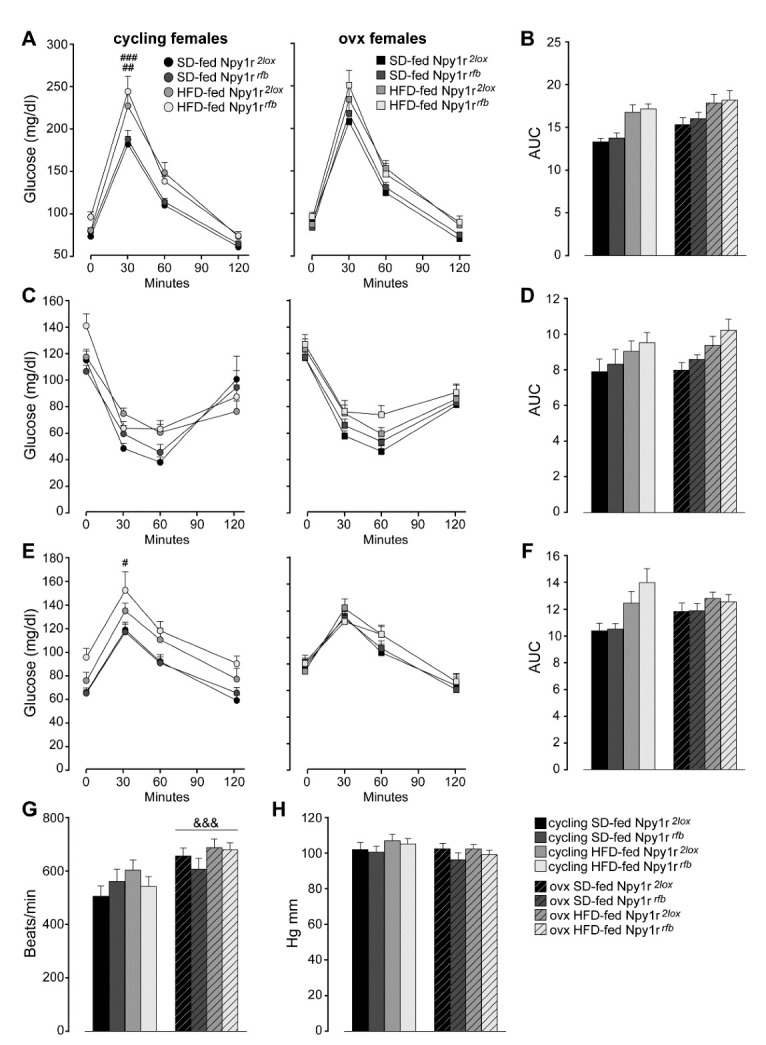
Glucose homeostasis and metabolism of standard diet (SD)- and high-fat diet (HFD)-fed cycling and ovx Npy1r*^2lox^* control and Npy1r*^rfb^* knockout female mice (**A**–**F**). (**A**) Glucose tolerance tests and (**B**) area under the curve (AUC) of SD- and HFD-fed cycling and ovx Npy1r*^2lox^* and Npy1r*^rfb^* females. Data are the mean ± SEM; n = 11–20 from 3 litters. ## *p* < 0.01 HFD-fed versus SD-fed cycling controls; ### *p* < 0.001 HFD-fed versus SD-fed cycling Npy1r*^2lox^* female mice. (**C**) Insulin tolerance test and (**D**) area under the curve (AUC) of SD- and HFD-fed cycling and ovx Npy1r*^2lox^* and Npy1r*^rfb^* females. Data are the mean ± SEM; n = 7–15 from 2 litters. (**E**) Pyruvate tolerance test and (**F**) area under the curve (AUC) of SD- and HFD-fed cycling and ovx Npy1r*^2lox^* and Npy1r*^rfb^* females. Data are the mean ± SEM; n = 7–16 from 2 litters. # *p* < 0.05 HFD-fed versus SD-fed cycling Npy1r*^rfb^* females. (**G**) Heart rate and (**H**) blood pressure of standard diet (SD)- and high-fat diet (HFD)-fed cycling and ovx Npy1r*^2lox^* control and Npy1r*^rfb^* knockout female mice. &&& *p* < 0.001 versus cycling females.

**Figure 4 ijms-23-06351-f004:**
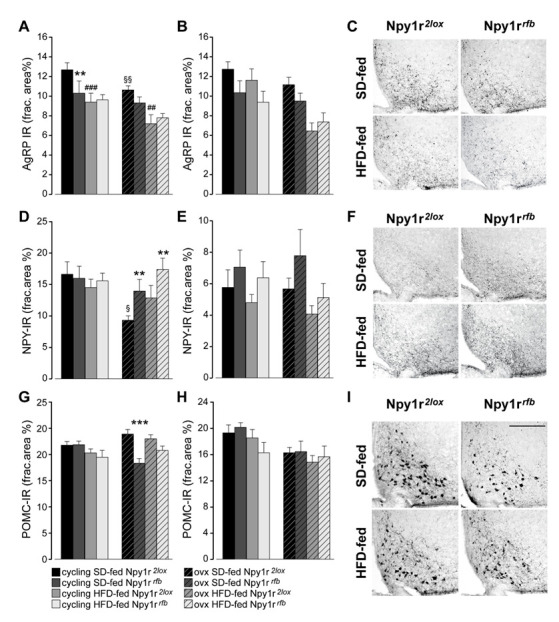
Expression of neuropeptides in the ARC and PVN of standard diet (SD)- and high-fat diet (HFD)-fed cycling and ovx Npy1r*^2lox^* control and Npy1r*^rfb^* knockout female mice. Agouti-related protein immunoreactivity (AgRP-IR) in the ARC (**A**) and in the PVN (**B**) of SD- and HFD-fed cycling and ovx Npy1r*^2lox^* and Npy1r*^rfb^* females. HFD and ovx significantly decreased AgRP expression in the ARC of female mice. SD-fed cycling Npy1r*^rfb^* conditional mutants showed a significant decrease of AgRP-IR in the ARC compared to SD-fed cycling control females. Data are the mean ± SEM; n = 8–9 (ARC) and n = 7–10 (PVN) from 2 litters. ** *p* < 0.01 and ### *p* < 0.001 versus SD-fed cycling Npy1r*^2lox^* females; §§ *p* < 0.01 versus SD-fed cycling Npy1r*^2lox^*; mice; ## *p* < 0.01 versus SD-fed ovx Npy1r*^2lox^* female mice. (**C**) Representative images of the ARC of ovx females expressing AgRP-IR immunoreactivity (scale bar: 150 μm). Neuropeptide Y immunoreactivity (NPY-IR) in the ARC (**D**) and in the PVN (**E**) of SD- and HFD-fed cycling and ovx Npy1r*^2lox^* and Npy1r*^rfb^* females. A significant decrease of NPY-IR was observed in the ARC of ovx Npy1r*^2lox^* mice compared with cycling Npy1r*^2lox^* mice. A significant increase of NPY-IR was observed in the ARC of ovariectomized Npy1r*^rfb^* mice compared with their control littermates on the same diet regimen. § *p* < 0.05 versus cycling Npy1r*^2lox^* females on the same diet regimen. ** *p* < 0.01 versus ovx Npy1r*^2lox^* on the same diet regimen. Data are the mean ± SEM; n = 5–8 from 2 litters. (**F**) Representative images of the ARC of ovx females expressing NPY-IR immunoreactivity (scale bar: 150 μm). Proopiomelanocortin immunoreactivity (POMC-IR) in the ARC (**G**) and in the PVN (**H**) of SD- and HFD-fed cycling and ovx Npy1r*^2lox^* and Npy1r*^rfb^* females. A significant decrease of POMC-IR was observed in the ARC of SD-fed ovx Npy1r*^rfb^* mice compared with SD-fed ovx Npy1r*^2lox^* mice. Data are the mean ± SEM; n = 7–11 from 2 litters. *** *p* < 0.001 versus SD-fed ovx Npy1r*^2lox^* mice. (**I**) Representative images of the ARC of ovx females expressing POMC-IR (scale bar: 150 μm).

**Figure 5 ijms-23-06351-f005:**
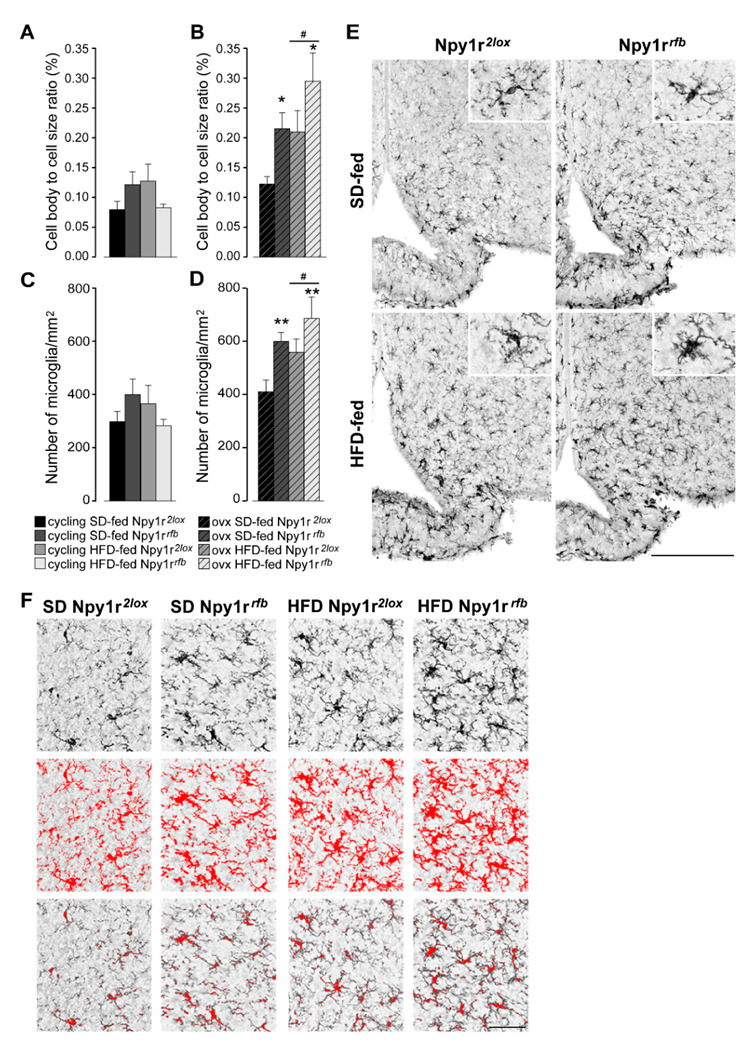
Effect of HFD and *Npy1r* gene deletion on microglial activation and microglia number. The cell body to cell size ratio (cb/c) of microglia (**A**,**B**) and the number of microglia/mm^2^ (**C**,**D**). HFD significantly increased cb/c (**B**) and microglia number (**D**) in the ARC of ovx females. Conditional inactivation of *Npy1r* gene increased cb/c (B) and microglia number (**D**) in the ARC of SD- and HFD-fed ovx mice. Data are the mean ± SEM; n = 4–6 from 2 litters. # *p* < 0.05 versus SD-fed ovx females. * *p* < 0.05 and ** *p* < 0.01 versus Npy1r*^2lox^* ovx females on the same diet regimen. (**E**) Representative images of the ARC of ovx females expressing ionized calcium-binding adaptor protein-1 (IBA-1) immunoreactivity (scale bar: 150 μm). (**F**) Image analysis used to quantify the morphological characteristics of microglia in Iba-1 staining in ovx females. Upper panels: the unprocessed pictures. Middle panels: pixels darker than the background are traced (red) to determine the total cell size of microglia. Lower panels: pixel-clusters that are above an applied staining threshold and size-filter are traced (red) to determine the total cell body size of all microglia, as well as the number of microglia.

**Figure 6 ijms-23-06351-f006:**
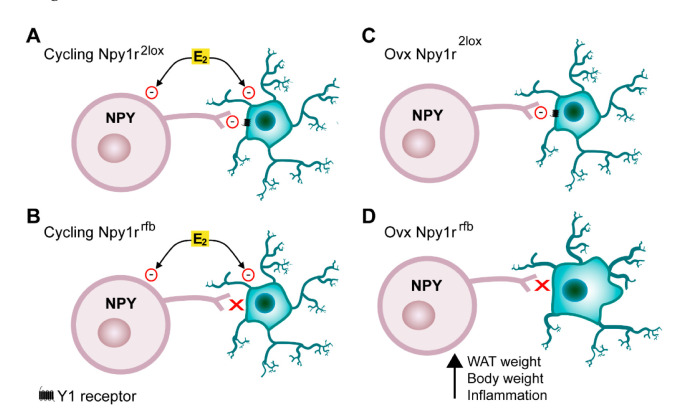
Interactions between estradiol and the NPY-Y1 receptor system in the ARC. (**A**) Estrogens (E2) directly inhibit NPY neurons. Microglia activation and proliferation is increased by HFD and inhibited by E2 and the NPY-Y1 receptor system. (**B**) In cycling females, the HFD-induced microglia activation is inhibited by E2 also in the absence of Y1 receptors on microglia. (**C**) In ovx Npy1r*^2lox^* females, NPY inhibits microglia through activation of Y1 receptors. (**D**) In Npy1r*^rfb^* ovx females, on both SD and HFD regimen, the deletion of Y1 receptors in microglia increases microglia activation and, in turn, body and WAT weight.

**Table 1 ijms-23-06351-t001:** Statistical analysis.

Measures	Female Mice	ANOVA	Variables	Interactions
Body weight	Cycling	2-way RM	Diet: F(1,101) = 80.54, *p* = 0.000	Days–diet: F(11,1111) = 22.06, *p* = 0.000
	ovx	2-way RM	Genotype: F(1,96) = 13.99 *p* = 0.000 Diet: F(1,96) = 91.04, *p* = 0.000	Days–diet–genotype: F(11,1056) = 4.41; *p* = 0.000 Days–genotype: F(11,1056) = 7.35 *p* = 0.000 Days–diet: F(11,1056) = 107.36, *p* = 0.000
Subcutaneous WAT	Cycling and ovx	3-way	Genotype: F(1,94) = 12.71, *p* = 0.001 Diet: F(1,94) = 279.77; *p* = 0.000 ovx: F(1,94) = 283.34; *p* = 0.000	Genotype–ovx: F(1,94) = 5.47; *p* = 0.022 ovx–diet: F(1,94) = 17.43; *p* = 0.000
Visceral WAT	Cycling and ovx	3-way	Diet: F(1,94) = 185.31; *p* = 0.000 ovx: F(1,94) = 116.32; *p* = 0.000	Genotype–diet: F(1,94) = 5.43; *p* = 0.022
Gonadic WAT	cycling and ovx	3-way	Diet: F(1,94) = 232.01; *p* = 0.000 ovx: F(1,94) = 33.14; *p* = 0.000	Diet–ovx: F(1,94) = 26.18; *p* = 0.000
Leptin	Cycling and ovx	3-way	Diet: F(1,117) = 107.80, *p* = 0.000 ovx: F(1,117) = 120.41, *p* = 0.000	
Spontaneous locomotor activity	Cycling and ovx	3-way	Diet: F(1,82) = 16.58; *p* = 0.000 ovx: F(1,82) = 41.42; *p* = 0.000	Genotype–ovx–diet: F(1,82) = 5.19; *p* = 0.025 ovx–diet: F(1,82) = 7.7905; *p* = 0.007
Cumulative energy intake	Cycling and ovx	3-way	Diet: F(1,24) = 52.50, *p* = 0.000	
9-day energy intake	Cycling and ovx	3-way RM	Diet: F(1,240) = 52.50, *p* = 0.000	
GTT	Cycling and ovx	3-way RM	Diet: F(1,112) = 25.62; *p* = 0.000 ovx: F(1,112) = 10.62; *p* = 0.002	Diet–hours: F(3,336) = 7.46; *p* = 0.000
GTT AUC	Cycling and ovx	3-way	Diet: F(1,112) = 26.38; *p* = 0.000 ovx: F(1,112) = 9.06; *p* = 0.003	
ITT	Cycling and ovx	3-way RM	Diet: F(1,80) = 9.74; *p* = 0.003	Diet–hours: F(3,240) = 6.08; *p* = 0.001
ITT AUC	Cycling and ovx	3-way	Diet: F(1,80) = 10.21; *p* = 0.002	
PTT	Cycling and ovx	3-way RM	Diet: F(1,85) = 19.04; *p* = 0.000	Diet–ovx: F(1,85) = 5.12; *p* = 0.026
PTT AUC	Cycling and ovx	3-way	Diet: F(1,85) = 19.23; *p* = 0.000	
Heart rate	Cycling and ovx	3-way	ovx: F(1,80) = 15.39; *p* = 0.000	
ARC AgRP-IR	Cycling and ovx	3-way	Diet: F(1,62) = 31.34, *p* = 0.000 ovx: F(1,62) = 21.1, *p* = 0.000	Genotype–diet: F(1,62) = 7.80, *p* = 0.007
PVN AgRP-IR	Cycling and ovx	3-way	Genotype: F(1,57) = 4.54, *p* = 0.037 Diet: F(1,57) = 12.13, *p* = 0.001 ovx: F(1,57) = 14.24, *p* = 0.000	
ARC NPY-IR	Cycling and ovx	3-way	Genotype: F(1,45) = 5.11, *p* = 0.029 ovx: F(1,45) = 4.82, *p* = 0.033	Genotype–ovariectomy: F(1,45) = 4.35, *p* = 0.043 ovx–diet: F(1,45) = 5.38, *p* = 0.025
ARC NPY-IR	ovx	2-way	Genotype: F(1,21) = 10.44, *p* = 0.004 Diet: F(1,21) = 6.11, *p* = 0.022	
ARC POMC-IR	Cycling and ovx	3-way	Genotype: F(1,68) = 15.00; *p* = 0.000	Genotype–ovx: F(1,68) = 6.86; *p* = 0.011 Diet–ovx: F(1,68) = 5.44; *p* = 0.023
PVN POMC-IR	Cycling and ovx	3-way	Diet: F(1,61) = 4.04; *p* = 0.049 ovx: F(1,61) = 10.48; *p* = 0.002	
Microglia cb/c	Cycling and ovx	3-way	Genotype: F(1,32) = 5.95; *p* = 0.021 Diet: F(1,32) = 5.98; *p* = 0.020 ovx: F(1,32) = 36.01; *p* = 0.000	Genotype–ovx: F(1,32) = 6.35; *p* = 0.017 Diet–ovx: F(1,32) = 4.80; *p* = 0.036
Microglia number	Cycling and ovx	3-way	Genotype: F(1,32) = 5.91; *p* = 0.021 ovx: F(1,32) = 44.80; *p* = 0.000	Genotype–ovx: F(1,32) = 4.61; *p* = 0.039 Diet–ovx: F(1,32) = 4.27; *p* = 0.047
Microglia cb/c	ovx	2-way	Genotype: F(1,15) = 7.23, *p* = 0.017 Diet: F(1,15) = 6.32, *p* = 0.024	
Microglia number	ovx	2-way	Genotype: F(1,15) = 8.80, *p* = 0.010 Diet: F(1,15) = 4.83, *p* = 0.044	

## Data Availability

Not applicable.

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
