# Peer review of "Hypothalamic NPY-Y1R Interacts with Gonadal Hormones in Protecting Female Mice against Obesity and Neuroinflammation"

_ijms, 2022, doi:10.3390/ijms23116351_

Round 1
Reviewer 1 Report
Overall, this is an interesting paper that evaluated the impact of NPY-Y1R on body weight. There are some minor concerns that are listed below:
1. Title - since the authors did not do an E2 replacement in the OVX animals, the title should be edited to replace estrogens with gonadal factors or not center on estrogens. The final paragraph of the introduction and the discussion section should explain more that it is not only estrogen involvement using the present experimental design.
2. It would be interesting to also measure brown back fat. If this information is available, the authors should include this data in the paper. The authors should also discuss the involvement/pathway for the brown fat.
Author Response
- Title - since the authors did not do an E2 replacement in the OVX animals, the title should be edited to replace estrogens with gonadal factors or not center on estrogens. The final paragraph of the introduction and the discussion section should explain more that it is not only estrogen involvement using the present experimental design.
From previous experiments we know that estrogen replacement is sufficient to prevent the ponderal weight induced by ovariectomy (Fontana et al. Endocrinology 2014, 155(6):2213-21); in addition, we have previously shown that there is a strong interaction between estrogen and NPY signalling (Fontana et al. Endocrinology 2014, 155(6):2213-21; Musso et al. Neuroendocrinology 2000, 72(6):360-67; Martini et al., Peptides 2011, 32(6):1330-34-4). Yet, progesterone receptor is present in the arcuate nucleus, and we did not carry out in the present study an estrogen replacement experiment; as a consequence, the suggestion is appropriate and the title (Hypothalamic NPY-Y1R interacts with gonadal hormones in protecting female mice against obesity and neuroinflammation) and text were appropriately amended by substituting the term estrogen with gonadal/ovarian hormones. On the other hand, we believe that estrogen is the best candidate since progesterone replacement does nor revert ovariectomy induced obesity (D Richard, Am J Physiol. 1986) and progesterone receptors are not expressed in microglia in adult mice (Villa, Endocrine Review, 2016), as discussed in the Discussion section, in bold, page 17, line 9 from the bottom.
- It would be interesting to also measure brown back fat. If this information is available, the authors should include this data in the paper. The authors should also discuss the involvement/pathway for the brown fat.
Unfortunately, the requested data are not available, but we do agree with the reviewer criticism and we acknowledge that specific insights on browning would be useful to further extend our knowledge on the impact of the cross-talk between hypothalamic NPY-Y1R system and gonadal steroids on metabolic parameter related with obesity. Accordingly, we are now planning a second paper aimed to better describe the peripheral effects of conditional inactivation of Npy1r gene and to elucidate the molecular mechanisms which can explain at least in part the effects here recorded. Thus, we thank the reviewer for the intriguing suggestion, which for sure we will take into consideration in the second study we have now planned to develop.

Reviewer 2 Report
The study is built on previous valuable results of the authors, which demonstrated that the downregulation of NPY1 receptors causes sex-dependent obesity. The manuscript addresses a relevant question as to how estrogen affects obesity induced by downregulation of NPY1 receptor. The performed experiments are well performed and documented. There are, however, some shortcomings in the paper, which should be addressed:
1. There is no description on the precise location of estrogen receptors through which it may exert its actions. Are they present in microglial cells in the arcuate nucleus?
2. The role of estrogen would be better recognized following estrogen replacement as well rather than only ovariectomy.
3. The manuscript neglects the possibility that target cells of NPY neurons may be involved in the observed changes. Is NPY1 receptor downregulated in the dorsomedial, paraventricular etc. hypothalamic nuclei?
4. Were microglial cells examined in other hypothalamic nuclei than the arcuate?
5. Is there any experimental evidence that NPY1 receptor is downregulated in the microglial cells of the arcuate nucleus?
Minor point:
It is not explained in the abstract why Lox mice are used. It should be written they are controls or deleted from the abstract.
Author Response
Answers to reviewers #2
- There is no description on the precise location of estrogen receptors through which it may exert its actions. Are they present in microglial cells in the arcuate nucleus?
Estrogen receptors are widely distributed in all cells of the arcuate nucleus. With regard to microglia, microglia expresses ERs (with a preponderance of ERα) and microglia cells in the CNS are patrolling the parenchyma thus move continuously attracted by inflammatory stimuli (see: Sex-Specific Features of Microglia from Adult Mice. Villa A, Gelosa P, Castiglioni L, Cimino M, Rizzi N, Pepe G, Lolli F, Marcello E, Sironi L, Vegeto E, Maggi A. Cell Rep. 2018 Jun 19;23(12):3501-3511. doi: 10.1016/j.celrep.2018.05.048)
- The role of estrogen would be better recognized following estrogen replacement as well rather than only ovariectomy.
The point is well taken therefore we amended the title (Hypothalamic NPY-Y1R interacts with gonadal hormones in protecting female mice against obesity and neuroinflammation) and text by substituting the term estrogen with gonadal/ovarian hormones. On the other hand, we believe that estrogen is the best candidate since progesterone replacement does nor revert ovariectomy induced obesity (D Richard, Am J Physiol. 1986) and progesterone receptors are not expressed in microglia in adult mice (Villa, Endocrine Review, 2016), as discussed in the Discussion section, in bold, page 17, line 9 from the bottom.
- The manuscript neglects the possibility that target cells of NPY neurons may be involved in the observed changes. Is NPY1 receptor downregulated in the dorsomedial, paraventricular etc. hypothalamic nuclei?
In the hypothalamus, Npy1r gene is significantly inactivated only in ARC and PVN of female Npy1rrfb mice, as previously reported (Bertocchi et al., Horm Behav. 2020, 125:104824) and now indicated in the Introduction Section, page 3, line 17 and 24. For this reason, we analysed NPY, AgRP and POMC expression in both of these nuclei. Our results clearly indicate the inflammatory response and the metabolic alterations observed in ovx Npy1rrfb females are associated with an increased expression of NPY and a significant decrease of POMC-IR in the ARC but not in the PVN. This suggests that ARC (where neuroinflammation was shown to alter POMC and NPY activity) is the hypothalamic nucleus where NPY-Y1R regulation of microglia and neuroinflammation occurs.
- Were microglial cells examined in other hypothalamic nuclei than the arcuate?
We measured microglia only in the ARC since it has been demonstrated that, in the hypothalamus, the diet- induced increase of neuroinflammation in male and in OVX female mice occurs mostly in the mediobasal hypothalamus (MBH), which contains the arcuate nucleus (ARC) and median eminence (ME). We are planning a second study to analyse the effect of HFD and OVX on microglia activation and number in other limbic regions involved in cognitive processes.
- Is there any experimental evidence that NPY1 receptor is downregulated in the microglial cells of the arcuate nucleus?
In Npy1rrfb mice, the conditional inactivation of the Npy1r gene occurs in all cells expressing the alpha-CaMKII promoter (Bertocchi et al., Proc Natl Acad Sci U S A 2011, 108(48):19395-400) which include forebrain glutamate positive cells and microglia. We agree with the reviewer that it could be interesting to quantify Npy1r gene expression in microglial cells, but this is a tricky experiment that, nevertheless, we are planning to perform in the next future.
Minor point:
It is not explained in the abstract why Lox mice are used. It should be written they are controls or deleted from the abstract.
We added to the abstract definition of conditional knockout Npy1rrfb and Npy1r2lox control mice (page 2, line 1).

Reviewer 3 Report
This is an interesting study investigating the interaction between Hypothalamic NPY-Y1R system with estrogens on metabolic parameter related with obesity and neuroinflammation in female mice. The authors using a conditional model to Npy1r gene in forebrain principal neurons show the consequences of Npy1r inactivation on metabolic parameters and leptin responses.
On overall the study is well written and has a good experimental rationale, however, is very limited on regards of the interpretation of the results and the conclusions that the authors want to obtain from them.
Major comments
- Conditional inactivation of Npy1r gene promoted gain weight in ovariectomized female mice with increasing of subcutaneous WAT. It correlates well with the results; however, the authors did not explore a possible target within the regulation of fatty acids synthesis that could be involved in these effects.
However, the authors measured circulating leptin levels, which can not justify the adiposity data. Did you check circulating ghrelin levels in these mice? Considering that ghrelin induces adiposity independent of orexigenic effects (PMID: 21543764). It would be very interesting for the whole phenotyping of these mice, to know if the obesogenic effect in this model is mediated by ghrelin.
- On the other hand, it is also curious that locomotor activity did not fit well with the gain weight because in ovariectomized mice fed a HFD with higher body weight did not show lower locomotor activity as would be expected. It is possible that recording of energy expenditure and body temperature could clear up this doubt, these data would be highly recommended if it is available to the authors.
- Any further speculation on the interaction of central estrogens receptors with NPY-Y1R system in the ARC and possible neurons involved should be included in the discussion.
Minor comments
- In some figures (Fig. 1A,1B, 2E…) the x-axis has not been properly identified and it is confusing. Please could you name if p66, p75 are days, or postnatal days? A classic x-axis defined in time as days or weeks, it would help us to better understanding of the data.
- Statistical analysis in the middle even at beginning of the paragraph, it makes very difficult for the reader to follow the text properly. Please, could statistical data be organized in tables outside the text? This issue would significantly facilitate the interpretation of the results for the reader.
- Methodological procedure to measure heart rate and blood pressure is missing in the manuscript.
Author Response
Answers to Reviewer #3
Major comments
- Conditional inactivation of Npy1r gene promoted gain weight in ovariectomized female mice with increasing of subcutaneous WAT. It correlates well with the results; however, the authors did not explore a possible target within the regulation of fatty acids synthesis that could be involved in these effects. However, the authors measured circulating leptin levels, which can not justify the adiposity data. Did you check circulating ghrelin levels in these mice? Considering that ghrelin induces adiposity independent of orexigenic effects (PMID: 21543764). It would be very interesting for the whole phenotyping of these mice, to know if the obesogenic effect in this model is mediated by ghrelin.
We really appreciate the reviewer’s valuable comments on the potential involvement of mechanisms linked to the regulation of fatty acids synthesis and ghrelin release in the described obesogenic effects. However, it would not be achievable to address these intriguing hypotheses in a reasonable time. Besides, the detailed investigation of the molecular events leading to the described peripheral effects goes beyond the main goal of the present paper, focused mainly on the study of the role of neuroinflammation in mediating the effects evoked by the interaction between ovarian hormones and NPY-Y1R transmission in the ARC. Nevertheless, we do recognize the importance of the relevant suggestions raised by the reviewers, and, accordingly, we have amended the text (Discussion section, page 16, line 28, in bold) and we are also planning a second study aimed to better elucidate the peripheral mechanisms underlying the effects here described.
- On the other hand, it is also curious that locomotor activity did not fit well with the gain weight because in ovariectomized mice fed a HFD with higher body weight did not show lower locomotor activity as would be expected. It is possible that recording of energy expenditure and body temperature could clear up this doubt, these data would be highly recommended if it is available to the authors.
We agree with the referee that it would have been interesting to measure energy expenditure and body temperature. Unfortunately, we did not record these parameters that therefore are not available. We thank the reviewer for the suggestion, which for sure we will take into consideration in future studies.
- Any further speculation on the interaction of central estrogens receptors with NPY-Y1R system in the ARC and possible neurons involved should be included in the discussion.
The possibility that other cross-talk mechanisms between estrogen and NPY may exist has been added to the Discussion section (page 18, line 6, in bold).
Minor comments
1. In some figures (Fig. 1A,1B, 2E…) the x-axis has not been properly identified and it is confusing. Please could you name if p66, p75 are days, or postnatal days? A classic x-axis defined in time as days or weeks, it would help us to better understanding of the data.
The figures 1A, 1B and 2E have been corrected according with the suggestion of the referee.
2. Statistical analysis in the middle even at beginning of the paragraph, it makes very difficult for the reader to follow the text properly. Please, could statistical data be organized in tables outside the text? This issue would significantly facilitate the interpretation of the results for the reader.
Statistical data have been organised in Table 1 (page 3, line 13 from the bottom; page 14-15)
3. Methodological procedure to measure heart rate and blood pressure is missing in the manuscript.
Heart rate and blood pressure measurements have been added to the Method Section (page 19, line 12 from the bottom).

Round 2
Reviewer 2 Report
The responses of the authors were extensive and satisfactory.
Author Response
Thank you
Reviewer 3 Report
Thanks for considering all my concerns and most of them have been resolved satisfactorily. However, in the comment 1, the authors included in the discussion the following sentence “On the other hand, we cannot exclude the involvement of other adipokines, such as ghrelin that induces adiposity independently of the orexigenic effect [46]”. Please, I recommend the authors not to define ghrelin as an adipokine because it may be confusing, since its expression is predominantly in the stomach and gut. For this reason, I strongly suggest the authors to name ghrelin as a gastric hormone instead of an adipokine.
Author Response
We added the term "gastric hormone" referring to ghrelin (page 16, line 20 from the bottom, in bold), according to the request of the reviewer.
